# *Plantago asiatica* Seed Extracts Alleviated Blood Pressure in Phase I–Spontaneous Hypertension Rats

**DOI:** 10.3390/molecules24091734

**Published:** 2019-05-04

**Authors:** Yaw-Syan Fu, Sheng-I Lue, Shiuan-Yea Lin, Chi-Lun Luo, Chi-Chen Chou, Ching-Feng Weng

**Affiliations:** 1Department of Biomedical Science and Environmental Biology, Kaohsiung Medical University, Kaohsiung 80708, Taiwan; m805004@kmu.edu.tw (Y.-S.F.); allen.luo1212@gmail.com (C.-L.L.); 2Department of Medical Research, Kaohsiung Medical University Hospital, Kaohsiung Medical University, Kaohsiung 80708, Taiwan; 3Center for Infectious Disease and Cancer Research, Kaohsiung Medical University, Kaohsiung 80708, Taiwan; 4Department of Physiology, Kaohsiung Medical University, Kaohsiung 80708, Taiwan; m655003@kmu.edu.tw; 5Department of Anatomy, Kaohsiung Medical University, Kaohsiung 80708, Taiwan; shiuanyea@yahoo.com.tw; 6Department of Life Science and Institute of Biotechnology, National Dong Hwa University, Hualien 97401, Taiwan; 410313054@gms.ndhu.edu.tw

**Keywords:** *Plantago asiatica*, hypertension, phase-1 hypertension

## Abstract

Arterial pressure of each new breeding spontaneous Phase-1 hypertension (P1-HT) rat was recorded for 5 min by intravascular femoral artery catheter that served as a reference value prior to treatment. In the acute antihypertensive test, 0.36 g/kg Bwt of *Plantago asiatica* seed extract (PSE) was administered, via gavage feeding, to P1-HT rats, and the arterial pressures were continuously recorded for 1 h. The acute antihypertensive effects of PSE on P1-HT rats appeared within 15 min after PSE administration and lasted over 1 h with systolic pressure decreased 31.5 mmHg and diastolic pressure decreased 18.5 mmHg. The systolic pressure decreased 28 mmHg and diastolic pressure decreased 16 mmHg in P1-HT rats when simultaneously compared with verapamil hydrochloride (reference drug), whereas there were no significant differences in the pretreated reference values of acute PSE treatment and the untreated control. In the chronic test, P1-HT rats received 0.36 g/kg Bwt day of PSE or equal volume of water for 4 weeks via oral gavage, and the lower blood pressure tendencies of chronic PSE treatment were also found when compared with the controls. The antihypertensive values of PSE were also confirmed in spontaneously hypertensive rats (SHRs). Oral administration with PSE can effectively moderate blood pressure within an hour, while taking PSE daily can control the severity of hypertension, suggesting PSE is a potentially antihypertensive herb.

## 1. Introduction

Hypertension is a multifactorial cardiovascular disorder that has a variety of genetic and environmental factors. Hypertension is a major risk factor for heart disease, heart attack, or stroke. Phase-1 hypertension (P1-HT) is a critical beginning stage of high blood pressure, which was referred to at the 2014 Joint National Committee 8 (JNC8), 2018 CSC (Chinese Society of Cardiology) hypertension guideline, and 2018 European Society of Cardiology/European Society of Hypertension (ESC). For decades, the clinical definition of systolic pressure >140 mmHg and diastolic pressure >90 mmHg were considered hypertension. Statistical results from the World Health Organization (WHO) and American Heart Association (AHA) have exposed that about one out of three U.S. adults have high blood pressure, and the WHO estimates more than one billion people suffer from hypertension [1]. Recently, the AHA 2017 guidelines redefined systolic pressure over 130 mmHg and diastolic pressure over 80 mmHg as hypertension. According to the AHA 2017 guidelines, about half of adults in the USA or Taiwan are considered to be hypertensive patients. In the same year, the American Academy of Family Physicians (AAFP) decided to not endorse the new hypertension guideline of AHA, and 2018 ESC and CSC hypertension guidelines maintained identical definitions of hypertension as JNC8. Remarkably, medicating hypertension is a public and urgent issue for health care. Therefore, researchers and drug companies gain insight into searching and developing more reliable and efficacious medicines obtained from natural sources to ameliorate the severity of hypertension.

Plantaginis semen is the dried ripe seed of *Plantago asiatica L.* and is a traditional medicine in China and other countries in Asia [2]. Traditionally, Plantaginis semen was applied for the treatment of several diseases. Plantaginis semen was reported e.g., the diuretic, anti-inflammation, and immunomodulation; and it has been used to treat blurred vision in Asia [2,3]. Streptozotocin-induced diabetic rats that were fed Plantaginis semen extract (PSE) showed an increase in the activities of superoxidase dismutase (SOD), catalase, and glutathione peroxidase in diabetic retinae [4]. In anti-obesity studies, PSE could improve lipid accumulation and hyperglycemia in high-fat diet-induced obese mice [5]. A recent study showed that PSE could inhibit the activities of xanthine oxidase, and it might be used in the reduction of hyperuricemia and the treatment of gout [6]. Aucubin is a major bioactive component that can be found in PSE [7], which has anti-inflammatory effects and protects the liver against damage induced by carbon tetrachloride or α-amanitin in mice and rats [8,9,10,11]. The seed coat of *P. asiatica* L. is mainly a polysaccharide, and there are many reports that have focused on discovering its anti-inflammatory [10], immunomodulatory [3,12,13], and anti-diabetic properties [4,14,15]. PSE polysaccharides could inhibit the activities of α-amylase and pancreatic lipase, they could inhibit protease activities, and they could slow down starch digestion and absorption to improve diabetic symptoms [16,17]. Furthermore, PSE polysaccharides could act on the mitogen-activated protein kinase (MAPK), nuclear factor NF-κB (NF-κB), and toll-like receptor 4 (TLR4) pathways to affect the maturation of bone marrow-derived dendritic cells [12,13,18,19], promote intestinal peristalsis and prevent constipation [16,20,21], and have antioxidant potency [22,23,24,25]. Moreover, the phenylpropanoid glycosides of PSE are also found to have strong antioxidant activity and significant immunoenhancing activity to stimulate the maturation of dendritic cells [3,4,13]. Although several medicinal values are reported for PSE, including diuretic, antioxidant, and anti-inflammatory effects, a rare study provided direct evidence that PSE has acute antihypertensive effects. The current study aims to administer PSE in both spontaneously P1-HT rats and spontaneously hypertensive rats (SHRs) to evaluate the antihypertensive effects of PSE after acute or chronic administrations.

## 2. Results

To determine the physiological condition of P1-HT rats, we measured all bred rats from neonatal to adult. Data showed that the body weights of neonatal P1-HT rats were 7.16 g in male and 7.17 g in female rats. The body weights of adult P1-HT rats 8 weeks old were 293 g in male and 191 g in female rats (Appendix A). This revealed that P1-HT rats retained the same body weight as their parental origin.

Next, to evaluate the baseline blood pressure of P1-HT rats, the blood pressure data of adult P1-HT (8 weeks old) rats showed the systolic pressure was 149 ± 11 mmHg, diastolic pressure was 91 ± 7 mmHg, mean arterial pressure was 115 ± 8 mmHg, pulse pressure was 58 ± 6 mmHg, and heart beat rate was 380 ± 50 BPM. Comparing the blood pressure data of two spontaneous hypertensive model rats, SHR rats (from Wistar rat, n = 5) and our new breeding P1-HT rats (from SD rat, *n* = 15), the blood pressure status of this new breed P1-HT rats could sustain the status of clinical phase I hypertension for more than one year (Figure 1).

Monitoring rat blood pressure confirmed that the systolic, diastolic, and mean arterial pressures of this spontaneous P1-HT rat were stably kept at 150 ± 20, 92 ± 30, and 115 ± 15 mmHg, respectively. The acute antihypertensive effects of this new breeding spontaneous P1-HT rat could appear within 15 min after PSE oral administration and last over 1 h. This acute antihypertensive effect could decrease the systolic pressure of P1-HT rats by 28 mmHg, and 16 mmHg for diastolic pressure when simultaneously compared with verapamil hydrochloride as a reference drug, while there were no significant differences in the reference value (5 min) of both acute PSE treated and the untreated control of P1-HT rat groups (Appendix A). This reducing tendency of blood pressure was also found after 4 weeks of PSE treatment when compared with the untreated control.

To determine the antihypertensive latency of *Plantago asiatica* seed extracts (PSEs), the acute antihypertensive effects of PSE (360 mg/kg, p.o., 1 h) on SHR rats (*n* = 5) were examined. After PSE treatment, the systolic and diastolic pressures decreased about 16.5 and 14.3 mmHg, respectively; the mean arterial pressure declined about 12.1 mmHg; and heart beat rate decreased about 100 BPM, but there was no significant effect on pulse pressure (Figure 2). The acute antihypertensive effects of PSE (360 mg/kg, p.o., 1 h) on P1-HT rats (*n* = 15) were examined next. The systolic and diastolic pressures decreased about 31.5 and 18.5 mmHg, respectively; the mean arterial and pulse pressures declined about 22.3 and 12.9 mmHg, respectively; and heart beat rate decreased about 100 BPM (Figure 3). These results revealed that the latent potency of PSE in controlling hypertension of P1-HT rats was more prominent than that of SHR. It suggested that P1-HT rats were a susceptible model for evaluating hypotensive drugs.

In investigating the acute (one time) and chronic antihypertensive effects of PSE (360 mg/kg, p.o., 4 weeks) on P1-HT rats (*n* = 5), data revealed that the antihypertensive effects of 4 weeks continuous oral administration of PSE had no significant difference from the first time PSE administration. This result showed that continuous oral administration did not abolish the antihypertensive effects by PSE treatment (Figure 4). Additionally, we also compared the antihypertensive effects of different types of clinical drugs and PSE on P1-HT rats (*n* = 5~6). In the PSE-treated group, systolic and diastolic pressures all decreased: 24.48 ± 6.23 (systolic) and 13.69 ± 3.90 mmHg (diastolic). After adjusting the range, there was no significant difference with propranolol (10.96 ± 10.55/7.16 ± 5.76 mmHg), bisoprolol (24.68 ± 10.01/18.64 ± 12.14 mmHg), and amlodipine/valsartan (23.59 ± 6.19/18.35 ± 8.87 mmHg) (Appendix A). This result indicated PSE possessed desirable antihypertensive efficacy compared to conventional hypotensive agents.

## 3. Discussion

In spontaneously hypertensive animals, such as SHR rats, the plasma levels of angiotensin II, endothelin 1 (ET-1), malonaldehyde (MDA), interleukin 6 (IL-6), tumor necrosis factor alpha (TNF-α), and NF-κB are upregulated, and nitric oxide (NO), nitric oxide synthases (NOS), and SOD levels are downregulated [26]. The oxidative stress and inflammatory response of spontaneously hypertensive animals might cause endothelial cell damage, aortic injury, and lead to hypertension. PSE treatment has anti-inflammatory and antioxidant effects by inhibiting activity of NF-κB, TNFα, cyclooxygenase-1 (COX-1), COX-2, and NO production [4,10,12]. Sustained administration of PSE may improve chronic high blood pressure.

The renin-angiotensin-aldosterone system is a major mechanism for blood pressure regulation. Previously, plantamajoside, acteoside, isoacteoside, and plantainoside were isolated from PSE, and their biological functions as angiotensin-converting enzyme (ACE) inhibitors were also demonstrated [27]. The activity of the inhibitory concentration 50% (IC_50_) on ACE of phenylethanoid glycosides was about 2.5 mM. The total amount of phenylethanoid glycosides in Plantaginis semen was less than 130 mg/kg [28], which indicated that the anti-hypotensive effect of PSE treatment was very weak through the inhibition of ACE activity by phenylethanoid glycosides. When compared with those studies, this application of PSE showed moderately attenuated hypertension in rats, which suggested that PSE might contain some other constituents rather than known compounds related to ACE inhibitors for exerting this action. It indicates that the bioavailability of isolated and identified compounds is low in regulation of blood pressure. Nevertheless, specific compounds of PSE for performing hypotensive action need to be further verified.

Diuretics are one of the major applications of Plantaginis semen in traditional medicine [2]. Na^+^/K^+^-ATPase activity is the major mechanism for fluid reabsorption in the kidneys, and PSE has strong inhibitory effects on renal Na^+^/K^+^-ATPase activity in horses [29]. The diuretic effect by inhibited Na^+^/K^+^-ATPase activity on renal tubular epithelium of PSE could decrease the total volume of blood by excluding water into the urine. Our results showed PSE treatment had an acute antihypertensive effect within 10 min and with little urine production. It is difficult to achieve acute antihypertensive effects by inhibiting Na^+^/K^+^-ATPase activity of renal tubule by PSE, but this inhibition of Na^+^/K^+^-ATPase activity may have the potential to be used as a long-term continuous blood pressure control.

Based on the results of this study, PSE treatment had the acute effects of lowering blood pressure and decreasing heart beat rate, which indicated PSE treatment acted on both the blood vessels to cause vasodilation and the heart to decrease cardiac output. This is the first evidence to report the effect of PSE treatment on cardiac output. However, the mechanical reaction of PSE treatment remains unclear and needs to be explored in future research.

## 4. Materials and Methods

### 4.1. Chemicals

The chemicals used in this study were all analytical grade and purchased from Sigma Aldrich (Merck KGaA, Darmstadt, Hesse, Germany). Preparations of *Plantago asiatica* seed extracts (PSEs) were described in our previous study [30]. Verapamil hydrochloride (5 mg/kg, Sigma, Oakville, ON, Canada), propranolol (1 mg/kg, Standard Chem & Pharm, Tainan, Taiwan), bisoprolol (0.5 mg/kg, Merck KGaA, Darmstadt, Hesse, Germany), and amlodipine/valsartan (0.5 mg/kg plus 8 mg/kg, Yungshin Pharm IND. Taichung, Taiwain) were obtained and all dissolved in water.

### 4.2. Animals

Two spontaneously hypertensive rats (SHRs) (from Wistar rat, *n* = 5; 10 weeks old) were obtained from the National Laboratory Animal Center (Taipei, Taiwan) and our new breeding P1-HT rats originated from Sprague Dawley (SD) (*n* = 15, 10 weeks old). Rats were kept in controlled environmental conditions at room temperature (22 ± 2 °C) and a humidity of 55% ± 10%. A 12 h light/dark (0600–1800) cycle was maintained throughout the study. The animals had free access to a commercial diet and were provided water ad libitum. Animal experimental protocols were followed as per the “Guide for the Care and Use of Laboratory Animals” of Kaohsiung Medical University approved by the Kaohsiung Medical University Institutional Animal Care and Use Committee (IACUC Approval No 102175).

### 4.3. Treatments

In this experiment, rats were anesthetized by 5% isoflurane (*v*/*v*). Anesthesia was maintained with an anesthesia mask supplied with 2% isoflurane (*v*/*v*) until the end of the experiment. The anesthesia mask was modified from a 10 mL plastic syringe. The experimental procedure was conducted as follows: the arterial pressures of each P1-HT rat were recorded for 5 min by an intravascular femoral artery catheter, which served as a reference value prior to treatment. In an acute antihypertensive effect test, P1-HT rats received, via gavage, 0.36 g/kg Bwt of PSE dissolved in distilled water or an equal volume of distilled water as a negative control after 5 min baseline monitoring. The arterial pressures were monitored continuously for 1 h. In general, an alternating profile of blood pressure was observed within 30 min and was sustained up to 60 min. Therefore, the duration of blood pressure changes within 1 h can be a typical pattern for determining the effect of treatment in these experiments. In the chronic antihypertensive effect test, P1-HT rats were orally fed, via gavage, and received 0.36 g/kg Bwt of PSE or equal volume of water every day for a 4 week period. Blood pressure recording procedures and the methods of chronic effect tests were the same as the acute effect tests throughout the 28-day PSE treatment regime. 

### 4.4. Statistical Analysis

All quantified results were shown as mean ± standard deviation (SD) of three independent experiments (*n* = 5 for three experiments). Significant analysis used a one-way ANOVA, followed by Student’s t-test to determine the difference in treatments compared to the untreated control.

## 5. Conclusions

In conclusion, this new breeding P1-HT model rat has a stable phase-1 hypertensive phenotype after reaching sexual maturity, and it is suitable for use in hypertensive studies in preclinical trials. Oral administration with PSE can effectively reduce blood pressure within 15 min and last more than 1 h, while chronic oral administration of PSE also found sustainable effects of hypotension, illustrating that taking PSE daily can control severity of hypertension.

## Figures and Tables

**Figure 1 molecules-24-01734-f001:**
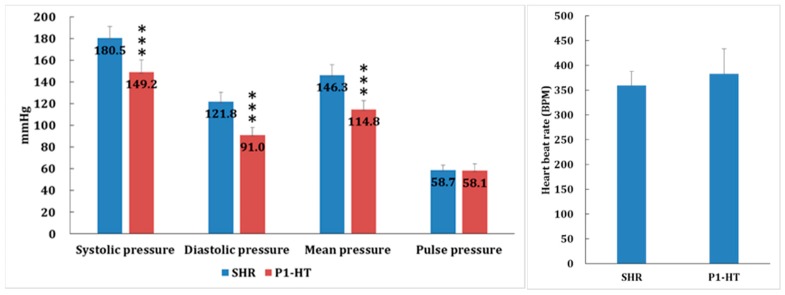
Blood pressure value comparisons of two spontaneous hypertensive rat (SHR) models: SHR rat (*n* = 5) and the new breeding P1-HT rat (*n* = 15). In the adult P1-HT rat (age 8 weeks) the systolic pressure is 149 ± 11 mmHg, diastolic pressure is 91 ± 7 mmHg, mean arterial pressure is 115 ± 8 mmHg, pulse pressure is 58 ± 6 mmHg, and heart beat rate is 380 ± 50 BPM. The blood pressure status of this new breed P1-HT rat could endure the tendency of clinical phase I hypertension for more than one year (*** *p* < 0.001).

**Figure 2 molecules-24-01734-f002:**
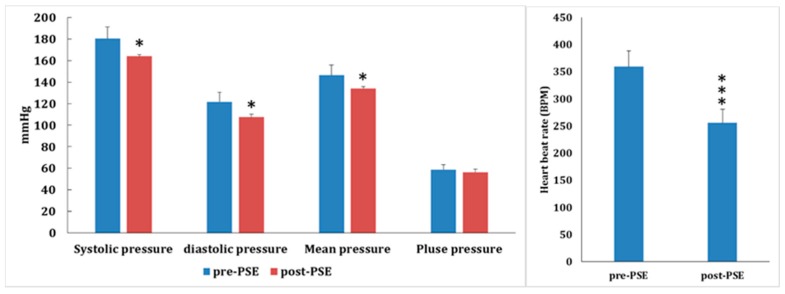
The acute antihypertensive effects of *Plantago asiatica* seed extracts (PSEs, 360 mg/kg, p.o., 1 h) on SHR rats (*n* = 5). Systolic pressure decreased about 16.5 mmHg, diastolic pressure decreased about 14.3 mmHg, the mean arterial pressure decreased about 12.1 mmHg, and heart beat rate was reduced about 100 BPM, but there was no significant effect on pulse pressure (* *p* < 0.05, *** *p* < 0.001).

**Figure 3 molecules-24-01734-f003:**
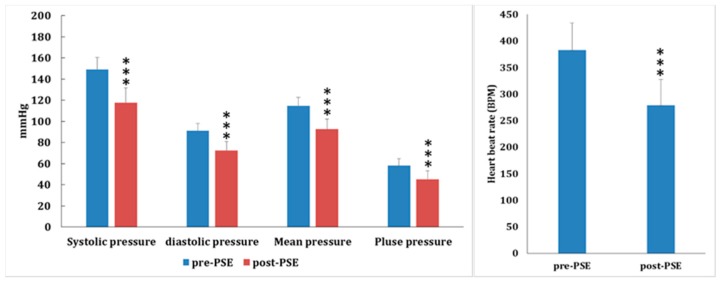
The acute antihypertensive effects of *Plantago asiatica* seed extracts (PSEs, 360 mg/kg, p.o., 1 h) on P1-HT rats (*n* = 15). Systolic pressure decreased about 31.5 mmHg, diastolic pressure decreased about 18.5 mmHg, the mean arterial pressure decreased about 22.3 mmHg, pulse pressure decreased about 12.9 mmHg, and heart beat rate decreased about 100 BPM (*** *p* < 0.001).

**Figure 4 molecules-24-01734-f004:**
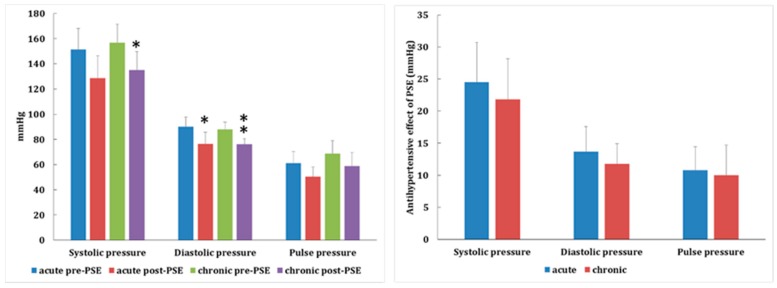
The one-time acute and chronic antihypertensive effects of *Plantago asiatica* seed extracts (PSEs, 360 mg/kg, p.o., 4 weeks) on P1-HT rats (*n* = 5). The antihypertensive effects after 4 weeks of continuous PSE oral administration were not significantly different from the first time PSE administration. This result showed that continuous oral administration did not negate the antihypertensive effects of PSE treatment (* *p* < 0.05, ** *p* < 0.01).

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
