# Peer review of "Plantago asiatica Seed Extracts Alleviated Blood Pressure in Phase I–Spontaneous Hypertension Rats"

_molecules, 2019, doi:10.3390/molecules24091734_

Round 1
Reviewer 1 Report
I consider that the research shows in the manuscript titled “Plantago asiatica seed extracts Alleviated Blood Pressure of Phase I–Spontaneous Hypertension Rats” is interesting and novel. In fact, the authors use a new hypertensive animal model to evaluate the antihypertensive properties of Plantago asiaica seed extracts. However, the explanation of the results is difficult to understand and the figures are not very explanatory. Moreover, it remains not clear if the PSE produces an antihypertensive effect after an acute administration since no differences were found between the blood pressure of rats administered water and the extract.
Other comments about the manuscript:
Abstract
Line 1. Define what P1-HT is
Introduction
-Please, provide information about what levels of blood pressure is considered to be phase 1-hypertension
-Authors indicate that “The main composition of P. asiatica L. seed coat is a polysaccharide, and there are many reports focused on the discovery of it as an anti-inflammatory [7], immunomodulation [3,8,9], and anti-diabetic [4,10,11].” Is the main composition a only polysaccharide or several?. I consider that this sentence should be followed by the sentence “Aucubin is a major bioactive component that can be found in PSE [22], which has anti-inflammatory effects and protects the liver against the damage induced by carbon tetrachloride or α-amanitin in mice and rats [7,23-25].”
-Authors indicate that “however, a rare study provided direct evidence that PSE has acute antihypertensive effects. This study administers PSE in both spontaneous P1-HT and SHR rats to evaluate the acute and chronic efficacy of antihypertension”. I consider that the expression acute or chronic antihypertensive effect should be change by antihypertensive effect of PSE after an acute or chronic administration. Authors evaluate the effect of an acute or chronic administration not the effect is not acute or chronic.
Results
-Authors indicate that “It reveals that P1-HT rats keep in the normal condition as a parental origin”. Does it mean that they maintain the same body weight?
-Please, change 91 ±7 mmHg for 91 ± 7 mmHg
-Authors indicate that “Comparing the blood pressure data of two spontaneous hypertensive model rats, SHR rats (from Wistar rat, n = 5) and our new breeding P1-HT rats (from SD rat, n = 15), this blood pressure status of this new breed P1-HT rats could maintain the range of clinical phase I hypertension for more than one year”. How do the authors evidence this fact? I think there is a mistake (a month instead of a year?)
-The authors indicate that the systolic blood pressure (SBP) of P1-HT rat 8 weeks old is 149 mmHg. However, according to the AHA 2017 guidelines, stage 1 hypertension is considered when SBP is between 130 and 139 mmHg. Please, explain why you consider this animal model is a good model to study P1-HT.
-Please, change 150±20 mmHg, 92±30 mmHg, and 115±15 mmHg for 150 ± 20 mmHg, 92 ± 30 mmHg, and 115 ± 15 mmHg
-The following paragraph is difficult to understand. Does PSE produce antihypertensive effect or not? “The acute antihypertensive effects of this new breeding spontaneous P1-HT rat could appear within 15 min after PSE oral administration and lasting more than 1 hr. This acute antihypertensive effect could decrease systolic pressure of P1-HT rat by 28 mmHg, and for diastolic pressure by 16 mmHg when simultaneously compared with verapamil as a reference drug, but there were no significant differences in the pretreated reference value of both acute PSE and the untreated control groups of P1-HT rat. The reducing tendency of blood pressure was also found in 4 weeks of PSE treatment as compared with the untreated control”. Please, include a graph which shows the blood pressure changes from 0 to 60 min post administration (water, verapamil and PSE treatments).
-“In PSE-treating group, systolic and diastolic pressure were all decreased 24.48 ± 6.23 (systolic) and 13.69 ± 3.90 mmHg (diastolic), respectively, which adjusting range was no significant difference with propranolol (10.96 ± 10.55/7.16 ± 5.76 mmHg), bisoprolol (24.68 ±10.01/18.64 ±12.14 mmHg) and amlodipine/valsartan (23.59 ± 6.19/18.35 ± 8.87 mmHg). This result indicated similar antihypertensive efficacy with conventional hypotensive agents”. Please, include a graph that show these results.
Discussion
I consider that discussion should be improved, including information about the effect of other seed extracts, polysaccharides included in the PSE. Moreover, provide information about bioavailability of the PSE compounds.
Materials and methods
-In the 4.1. section is indicated that “Natural compounds were dissolved in corn oil”, however, in the section 4.2, they explain that PSE was dissolved in water to be administered to animals. Please, provide more information about this item.
-Please, define SD (Sprague dawley rats?)
-Please provide information about the age of the SHR
-The authors used water and verapamil as negative and positive controls, however this information is not described in this section. Please, provide information about this item (volume, concentration, etc). Moreover, they tested the effect of several drugs propranolol, bisoprolol and amlodipine. Information about this experiment is not provided even though it is not clear if the experiment is after a chronic or acute administration.
-Why is 1 hour post-administration is evaluated?
Author Response
Manuscript ID: molecules-487863
We greatly appreciate the helpful comments of the reviewers. We have made the necessary changes, which we believe substantially improved the manuscript. In response to the reviewers’ comments, we have added new sentences and edited existing sentences in the revised version according the comments. All of the changes are yellow highlighted in the revised manuscript. Our point-by-point responses are set out below.
Responses to specific comments from reviewer #1
I consider that the research shows in the manuscript titled “Plantago asiatica seed extracts Alleviated Blood Pressure of Phase I–Spontaneous Hypertension Rats” is interesting and novel. In fact, the authors use a new hypertensive animal model to evaluate the antihypertensive properties of Plantago asiaica seed extracts.
Q. However, the explanation of the results is difficult to understand and the figures are not very explanatory. Moreover, it remains not clear if the PSE produces an antihypertensive effect after an acute administration since no differences were found between the blood pressure of rats administered water and the extract.
A: The authors thanks the value comments of eminent reviewer, we will modify all necessary
points of comments.
Other comments about the manuscript:
Abstract
Q: Line 1. Define what P1-HT is
A: Many thanks for reviewer’s comment, we will define P1-HT as you can see in the yellow highlights.
Introduction
Q: -Please, provide information about what levels of blood pressure is considered to be phase 1-hypertension
A: Many thanks for reviewer’s comment, we will describe P1-HT in detail, please see in yellow highlights.
Q:-Authors immunomodulation [3,8,9], and anti-diabetic [4,10,11].” Is the main composition a only polysaccharide or several?. I consider that this sentence should be followed by the sentence “Aucubin is a major bioactive component that can be found in PSE [22], which has anti-inflammatory effects and protects the liver against the damage induced by carbon tetrachloride or α-amanitin in mice and rats [7,23-25].”
A: We did move all sentences to appropriate site as reviewer’s suggestion and the citations are also renumbered.
Q: -Authors indicate that “however, a rare study provided direct evidence that PSE has acute antihypertensive effects. This study administers PSE in both spontaneous P1-HT and SHR rats to evaluate the acute and chronic efficacy of antihypertension”. I consider that the expression acute or chronic antihypertensive effect should be change by antihypertensive effect of PSE after an acute or chronic administration. Authors evaluate the effect of an acute or chronic administration not the effect is not acute or chronic.
A: Many thanks for reviewer’s comment, we did change as reviewer’s suggestion.
Results
Q: -Authors indicate that “It reveals that P1-HT rats keep in the normal condition as a parental origin” Does it mean that they maintain the same body weight?
A: Many thanks for reviewer’s comment, we have changed as you can see in yellow highlights.
Q:-Please, change 91 ±7 mmHg for 91 ± 7 mmHg
A: Many thanks for reviewer’s comment, we have changed the typos.
Q:-Authors indicate that “Comparing the blood pressure data of two spontaneous hypertensive model rats, SHR rats (from Wistar rat, n = 5) and our new breeding P1-HT rats (from SD rat, n = 15), this blood pressure status of this new breed P1-HT rats could maintain the range of clinical phase I hypertension for more than one year”. How do the authors evidence this fact? I think there is a mistake (a month instead of a year?)
A: Many thanks for reviewer’s comment, in fact the new breed P1-HT rats can sustain the hypertensive stats for more than one year.
Q:-The authors indicate that the systolic blood pressure (SBP) of P1-HT rat 8 weeks old is 149 mmHg. However, according to the AHA 2017 guidelines, stage 1 hypertension is considered when SBP is between 130 and 139 mmHg. Please, explain why you consider this animal model is a good model to study P1-HT.
A: Many thanks for reviewer’s comment, we consider 2018 ESC/ESH hypertension guideline, 2018 CSC (Chinese Society of Cardiology) hypertension guideline, and 2017 the decision of AAFP, most maintained the same definition of hypertension as JNC8. Accordingly, we still keep the hypertensive rats defined as P1-HT.
Q: -Please, change 150±20 mmHg, 92±30 mmHg, and 115±15 mmHg for 150 ± 20 mmHg, 92 ± 30 mmHg, and 115 ± 15 mmHg
A: Many thanks for reviewer’s comment, we have changed these typos.
Q: -The following paragraph is difficult to understand. Does PSE produce antihypertensive effect or not? “The acute antihypertensive effects of this new breeding spontaneous P1-HT rat could appear within 15 min after PSE oral administration and lasting more than 1 hr. This acute antihypertensive effect could decrease systolic pressure of P1-HT rat by 28 mmHg, and for diastolic pressure by 16 mmHg when simultaneously compared with verapamil as a reference drug, but there were no significant differences in the pretreated reference value of both acute PSE and the untreated control groups of P1-HT rat. The reducing tendency of blood pressure was also found in 4 weeks of PSE treatment as compared with the untreated control”. Please, include a graph which shows the blood pressure changes from 0 to 60 min post administration (water, verapamil and PSE treatments).
A: Many thanks for reviewer’s comment, we have modified “while there were no significant differences in the reference value (5 min) of both acute PSE treated and the untreated control of P1-HT rat groups.” And we have also provided a new supplement Fig. S2.
Q: -“In PSE-treating group, systolic and diastolic pressure were all decreased 24.48 ± 6.23 (systolic) and 13.69 ± 3.90 mmHg (diastolic), respectively, which adjusting range was no significant difference with propranolol (10.96 ± 10.55/7.16 ± 5.76 mmHg), bisoprolol (24.68 ±10.01/18.64 ±12.14 mmHg) and amlodipine/valsartan (23.59 ± 6.19/18.35 ± 8.87 mmHg). This result indicated similar antihypertensive efficacy with conventional hypotensive agents”. Please, include a graph that shows these results.
A: Many thanks for reviewer’s comment, we have changed and provided a new supplement Figure S3.
Discussion
I consider that discussion should be improved, including information about the effect of other seed extracts, polysaccharides included in the PSE. Moreover, provide information about bioavailability of the PSE compounds.
A: Many thanks for reviewer’s comment, we have changed in second paragraph.
Materials and methods
Q: -In the 4.1. section is indicated that “Natural compounds were dissolved in corn oil”, however, in the section 4.2, they explain that PSE was dissolved in water to be administered to animals. Please, provide more information about this item.
A: Many thanks for reviewer’s comment, we have changed the mistake because PSE was indeed dissolved in water.
Q: -Please, define SD (Sprague dawley rats?)
A: Many thanks for reviewer’s comment, we have given full name for SD.
Q:-Please provide information about the age of the SHR
A: Many thanks for reviewer’s comment, we have added the age for SHR rats.
Q: -The authors used water and verapamil as negative and positive controls, however this information is not described in this section. Please, provide information about this item (volume, concentration, etc). Moreover, they tested the effect of several drugs propranolol, bisoprolol and amlodipine. Information about this experiment is not provided even though it is not clear if the experiment is after a chronic or acute administration.
A: Many thanks for reviewer’s comment, we have added all information about tested drugs and provided a new supplement Figure S2.
Q: -Why is 1 hour post-administration is evaluated?
A: Many thanks for reviewer’s comment. The experimental procedure was conducted as follow: the arterial pressures of each P1-HT rats were recorded for 5 min by intravascular femoral artery catheter served as a reference value prior to treatment. In acute antihypertensive effect test, P1-HT rat gavage received 0.36 g/kg Bwt of PSE by dissolved in distilled water or equal volume of distilled water as a negative control and the arterial pressures were monitored continuously for 1 hr. In general, the alternating profile of blood pressure was observed within 30 min and stably sustained the observed trend up to 60 min. Therefore, the duration of blood pressure changes with 1 hr can be a typical pattern for determining the effect of treatment in these experiments.
Again, we greatly appreciate all of your helpful and constructive comments. We look forward to your reply in your convenient time.
Sincerely yours,
Ching-Feng Weng, Ph.D., Professor
Department of Life Science and Institute of Biotechnology
National Dong Hwa University
Shoufeng, Hualien 97401, Taiwan
Phone: +886-3-890-3637

Reviewer 2 Report
In this manuscript (ID: molecules-487863), titled “Plantago asiatica seed extracts alleviated blood pressure of phase I-spontaneous hypertension rats”, the authors, Fu et al, detected the effect of PSE on the blood pressure and heart rate of P1-HT and SH rats. Their results demonstrate that oral administration of PSE significantly reduces the blood pressure and heart rate in both P1-HT and SH rats. They conclude that PSE is an antihypertensive herb. The results from this study on hypertension is relatively novel. However, there are several major concerns needed to be addressed, which are listed in the following:
1) The major concern is that the current study is superficial without any investigation on the possible mechanism involved in the antihypertension effect. Although the vasodilation was proposed, there is no evidence provided.
2) The Sprague Dawley rats are normotensive. How did you breed them into a hypertensive model? Why this hypertensive animal model is unique as compared with other hypertensive model? Where did you purchase these SD rats in your study?
3) Please provide detail information regarding oral administration of PSE in the rats connecting both arterial catheter and isoflurane tube, because this procedure could affect the blood pressure recording. It is almost impossible for the chronic experiment.
4) The PSE was dissolved in corn oil. Therefore, the corn oil should be used as control instead of water.
5) The time-dependent curve of blood pressure and heart rate should be provided in order to analyze the possible mechanism involved.
6) In statistical analysis, please explain “three-independent experiments”, meaning 3 rats or 3 recordings in 1 rat? Please delete the last sentence about Excel because it is not related to the result.
7) The manuscript language and grammar should be carefully checked and edited.
Author Response
Manuscript ID: molecules-487863
We greatly appreciate the helpful comments of the reviewers. We have made the necessary changes, which we believe substantially improved the manuscript. In response to the reviewers’ comments, we have added new sentences and edited existing sentences in the revised version according the comments. All of the changes are yellow highlighted in the revised manuscript. Our point-by-point responses are set out below.
Responses to specific comments from reviewer #2
In this manuscript (ID: molecules-487863), titled “Plantago asiatica seed extracts alleviated blood pressure of phase I-spontaneous hypertension rats”, the authors, Fu et al, detected the effect of PSE on the blood pressure and heart rate of P1-HT and SH rats. Their results demonstrate that oral administration of PSE significantly reduces the blood pressure and heart rate in both P1-HT and SH rats. They conclude that PSE is an antihypertensive herb. The results from this study on hypertension is relatively novel. However, there are several major concerns needed to be addressed, which are listed in the following:
Q: 1) The major concern is that the current study is superficial without any investigation on the possible mechanism involved in the antihypertension effect. Although the vasodilation was proposed, there is no evidence provided.
A: The authors thanks the value comments of eminent reviewer, we will modify all necessary points of comments.
Q: 2) The Sprague Dawley rats are normotensive. How did you breed them into a hypertensive model? Why this hypertensive animal model is unique as compared with other hypertensive model? Where did you purchase these SD rats in your study?
A: Many thanks for reviewer’s comment, we have originally used SD rats as a normal BP control in previous study. Unfortunately, we all failed for several times and latterly we decided to raise our own strain SD rats. After backcross and inbreed, we can successfully breed this new P1-HT rats at least 18 generations. We have checked their BP condition with comparison of SHR and indeed they show the stable hypertension.
Q: 3) Please provide detail information regarding oral administration of PSE in the rats connecting both arterial catheter and isoflurane tube, because this procedure could affect the blood pressure recording. It is almost impossible for the chronic experiment.
A: Many thanks for reviewer’s comment. The experimental procedure was conducted as follow: the arterial pressures of each P1-HT rats were recorded for 5 min by intravascular femoral artery catheter served as a reference value prior to treatment. In acute antihypertensive effect test, P1-HT rat gavage received 0.36 g/kg Bwt of PSE by dissolved in distilled water or equal volume of distilled water as a negative control and the arterial pressures were monitored continuously for 1 hr. In general, the alternating profile of blood pressure was observed within 30 min and stably sustained the observed trend up to 60 min. Therefore, the duration of blood pressure changes with 1 hr can be a typical pattern for determining the effect of treatment in these experiments.
Q: 4) The PSE was dissolved in corn oil. Therefore, the corn oil should be used as control instead of water.
A: Many thanks for reviewer’s comment, we made mistake and in fact the PSE was dissolved in water.
Q: 5) The time-dependent curve of blood pressure and heart rate should be provided in order to analyze the possible mechanism involved.
A: Many thanks for reviewer’s comment, we provided the time-dependent cure of blood pressure and heart rate for reviewer reference.
Q: 6) In statistical analysis, please explain “three-independent experiments”, meaning 3 rats or 3 recordings in 1 rat? Please delete the last sentence about Excel because it is not related to the result.
A: Many thanks for reviewer’s comment, the experiments ran for three times with 5 rats in each time. And we have deleted this sentence.
Q: 7) The manuscript language and grammar should be carefully checked and edited.
A: Many thanks for reviewer’s comment, we have rigorously checked grammatic errors.
Again, we greatly appreciate all of your helpful and constructive comments. We look forward to your reply in your convenient time.
Sincerely yours,
Ching-Feng Weng, Ph.D., Professor
Department of Life Science and Institute of Biotechnology
National Dong Hwa University
Shoufeng, Hualien 97401, Taiwan
Phone: +886-3-890-3637

Round 2
Reviewer 2 Report
The manuscript has been improved and I do not have other concerns.